# Highly Accurate Pose Estimation as a Reference for Autonomous Vehicles in Near-Range Scenarios

**Ursula Kälin [1], Louis Staffa [2], David Eugen Grimm [1,](https://orcid.org)* and Axel Wendt [2]**

[1] Institute Geomatics, School of Architecture, Civil Engineering and Geomatics, University of Applied Sciences and Arts Northwestern Switzerland, 4132 Muttenz, Switzerland; ursula.kaelin@fhnw.ch

[2] Robert Bosch GmbH, 70049 Stuttgart, Germany; Louis.Staffa@de.bosch.com (L.S.); Axel.Wendt@de.bosch.com (A.W.)

\* Correspondence: david.grimm@fhnw.ch

**Abstract:** To validate the accuracy and reliability of onboard sensors for object detection and localization for driver assistance, as well as autonomous driving applications under realistic conditions (indoors and outdoors), a novel tracking system is presented. This tracking system is developed to determine the position and orientation of a slow-moving vehicle during test maneuvers within a reference environment (e.g., car during parking maneuvers), independent of the onboard sensors. One requirement is a 6 degree of freedom (DoF) pose with position uncertainty below 5 mm ($3\sigma$), orientation uncertainty below 0.3° ($3\sigma$), at a frequency higher than 20 Hz, and with a latency smaller than 500 ms. To compare the results from the reference system with the vehicle's onboard system, synchronization via a Precision Time Protocol (PTP) and system interoperability to a robot operating system (ROS) are achieved. The developed system combines motion capture cameras mounted in a 360° panorama view setup on the vehicle, measuring retroreflective markers distributed over the test site with known coordinates, while robotic total stations measure a prism on the vehicle. A point cloud of the test site serves as a digital twin of the environment, in which the movement of the vehicle is visualized. The results have shown that the fused measurements of these sensors complement each other, so that the accuracy requirements for the 6 DoF pose can be met while allowing a flexible installation in different environments.

**Keywords:** motion capture camera; robotic total station; autonomous vehicle; 6 DoF pose estimation; accuracy

## 1. Introduction

Autonomous driving algorithms use different integrated sensors (camera, lidar, radar, etc.) whose outputs are fused to provide information to control the vehicle. For the safe release of such autonomous vehicles, these sensors must fulfill strict requirements in terms of spatial and detection accuracy. Environmental conditions such as weather effects, external light sources and other traffic participants can influence the sensors' performance, potentially impacting behavior in real traffic situations. Therefore, any argument for the release of an autonomous vehicle must be based on data that reflect the conditions in real traffic.

Methods that can be used to achieve sufficient testing of the sensors under the required conditions are statistical analyses of drives in real traffic, as well as scenario-based simulations. These approaches cover most requirements and can be used for release arguments, but they can also be very costly and favor scenarios that occur frequently during regular traffic.

A special set of scenarios that autonomous driving vehicles must handle involve near-range scenarios such as maneuvering in and out of parking spots. These situations usually only occur a few times per drive and at low speeds, although they require higher accuracy in terms of localization and object detection compared to flowing traffic. While

an autonomous vehicle can keep a safe distance of more than a meter from any traffic participant at any time in regular traffic, a tight parking spot might require maneuvering with only 20 cm of space or less on either side of the vehicle.

A novel, independent measuring system is necessary in order to provide proof that the autonomous vehicle's sensors fulfill the necessary requirements in terms of accuracy and reliability in realistic near-range scenarios. Due to the mostly static environment and low driving speeds while parking indoors and outdoors, typical references such as global navigation satellite system (GNSS) devices on target vehicles and onboard inertial measurement units (IMU) can only be used to a limited extent due to missing satellite reception in indoor scenarios (GNSS) and the drift behavior of the IMU. Additional cameras and visual SLAM could stabilize the drift behavior during GNSS outages, as presented in [1]. Their positioning system works in real-time, is low-cost, and can determine the 6DOF pose of the vehicle. Although the method uses information from the sensors in the vehicle to improve the position accuracy, the accuracy requirements could not be met here. In addition, this means that the system is no longer independent of the onboard sensors.

The integration of this tracking system into the existing onboard sensor environment of the vehicle leads to additional requirements. This includes the establishment of a link between the reference frames as well as the communication between the components. The accuracy requirements are specified to be one order of magnitude higher than the tested system at a similar message frequency.

Despite the large range of systems used for determining positions or poses, there is no standard system that meets all of the requirements modeled in this publication. Thus, a novel combination of external sensors was developed in this research to track the precise position and orientation of the vehicle under testing in a local reference system. The requirement of a 6 degree of freedom (DoF) pose with a position uncertainty below 5 mm (3σ), orientation uncertainty below 0.3° (3σ), at a frequency higher than 20 Hz, and with a latency smaller than 500 ms was achieved using known industrial metrology methods. However, the system still needs to work outside of laboratory conditions, which will involve varying meteorological and illumination conditions and a measuring range of up to 100 m. This section provides an overview of the measurement systems considered.

Robotic total stations or multistations determine the coordinates of a target (e.g., prism) by measuring horizontal and vertical angles and distances. Thanks to automated target aiming, it is possible to automate the tracking and measurement of a 360° prism mounted on a vehicle. The Leica MultiStation MS60 has a specified accuracy for angle measurements of 1″ (0.3 mgon) and distance measurements of 1 mm (+1.5 ppm) [2]. Nevertheless, the structure of the 360° prism introduces additional systematic deviations dependent on its orientation. A previous study [3] showed that for a GRZ122 prism, cyclic errors of more than 1 mm can be detected, along with larger deviations for specific prism orientations. The limiting factor of the maximum achievable measurement rate of 20 Hz is the performance of the electronic distance measurement (EDM), as described by [4].

Using the Leica GeoCOM communication interface allows for the integration of the MS60 in a sensor network and the control of the instrument from an external program. However, this interface does not have a native synchronization option based on standardized network-based synchronization protocols. Investigations in [5,6] regarding total stations showed that the synchronization quality has more significance in terms of the spatiotemporal accuracy with faster movement of the tracked prism.

In order to estimate the 6 DoF pose, it is necessary to obtain simultaneous measurements of three prisms on a vehicle. Assuming possible measurement deviations of 4.3 mm, 5.0 mm, and 2.9 mm in longitudinal, lateral, and vertical directions, respectively, the required baseline length to meet the orientation accuracy can be calculated using a variance propagation with partial derivatives of $\tan^{-1}\left(\frac{x_2 - x_1}{y_2 - y_1}\right)$. The resulting baseline length of a minimum of 3.8 m is not feasible for automotive applications. By using a Laser Tracker instead of a total station, the accuracy can be increased significantly. The Leica AT960 absolute tracker, for example, has a specified measurement performance well below 1 mm [7].

With the help of a built-in camera, it is possible to determine the 6 DoF pose of a specific probe (T-Mac). However, laser trackers typically used in manufacturing are dependent on stable environmental conditions (vibration, air temperature, air pressure) and have a limited measuring range, usually of less than 50 m. In addition, a cable connection between the laser tracker and T-Mac is required in most cases and the automated target recognition is not reliable in sunlight. These constraints make this instrument unusable for the task at hand.

Radar-based systems use microwaves with wavelengths of between 1 mm and 1 cm. A distance measurement with radar can be determined over the time of flight. Radar systems used for navigational purposes do not achieve the high accuracy requirements set for this paper [8]. To carry out high-precision surveying work—which is typically used for monitoring rockslides or dams—changes in distance are perceived with interferometry. The static setup of the radars does not allow an object to be tracked in a partly obstructed environment and is more useful for surveying an area rather than a single moving target. So-called laser–radar systems [9] are used in industrial metrology for surface inspection. However, they are not able to track objects, meaning they are not suitable for our task.

The iGPS (indoor GPS) system from 7D Kinematic Metrology (earlier Nikon Metrology) is a local positioning system consisting of several stationary infrared transmitters that emit two inclined beams in a rotating motion. From the time differences of the detected light signals on a sensor, it is possible to calculate the azimuth and elevation of the transmitter. By measuring the directions to multiple transmitters, it is feasible to calculate the position. Based on the accuracy assessment in [10], velocities of up to 3 m/s with deviations of up to 1.3 mm can be successfully tracked. The time offset must be considered for spatiotemporal applications, which varies with the speed of the sensor. The system latency is around 300 ms [11] and the measurements are not accessible for real-time streaming.

Motion capture systems estimate the positions of retroreflective markers using a stereo system of active cameras surveying the measurement volumes at high frequencies, e.g., up to 420 Hz for a Vicon Vantage V5. The cameras are sensitive to specific wavelengths, which either come from retroreflective markers reflecting illuminations from the cameras or from active luminescent markers. Such motion capture systems are mainly used for movement analysis in movie productions, sports, and medical and robotic applications [12].

The positioning performance study presented in [13] investigated a system with eight Vicon T40S cameras monitoring a rotating arm from distances of <5 m. The results showed high accuracy with errors in the sub-millimeter range. This application can be problematic due to the high number of cameras required for bigger measurement volumes, as well as the possible disturbance signals that are caused by sunlight reflections on surfaces.

Light detection and ranging (LiDAR) sensors scan the surrounding areas at high frequencies and create continuously updated 3D point clouds. With simultaneous localization and mapping (SLAM) methods, self-localization of the vehicle can be achieved. Specifications from LiDAR manufacturers indicate measurement accuracies of 2 cm and up [14]. Various studies [15–18] have showed that with this technique, it is possible to reach root mean square error (RMSE) values in the range of decimeters for position measurements and an RMSE value of 0.1° for the heading measurement. The results primarily depend on the precision and density of the available 3D data. The computationally intensive algorithms make it hard to implement this technique for real-time applications with high accuracy requirements. In addition, this technology is identical to one of the vehicle sensors tested in this study, hindering the possibility of an independent assessment.

An inertial measurement unit (IMU) with built-in accelerometers and gyroscopes measures the linear accelerations and rotation rates. To derive changes in position, double integration over time is necessary. This measurement principle returns data at high rates ranging from 100 Hz up to kHz [19,20], but with significant drift. Therefore, an IMU could complement other measurement systems with frequent measurements but would require correction values to control the drift.

## 2. Materials and Methods

The proposed tracking system consists of a novel combination of motion capture cameras (Vicon Vantage V5) and robotic MultiStation instruments (Leica MS60).

Instead of the classical motion capture approach, involving the monitoring of the motions of marker-tagged objects from multiple angles with static cameras, the principle is inverted and the cameras are placed with a panoramic view on top of the moving object (Figure 1) to detect static markers distributed over the area.

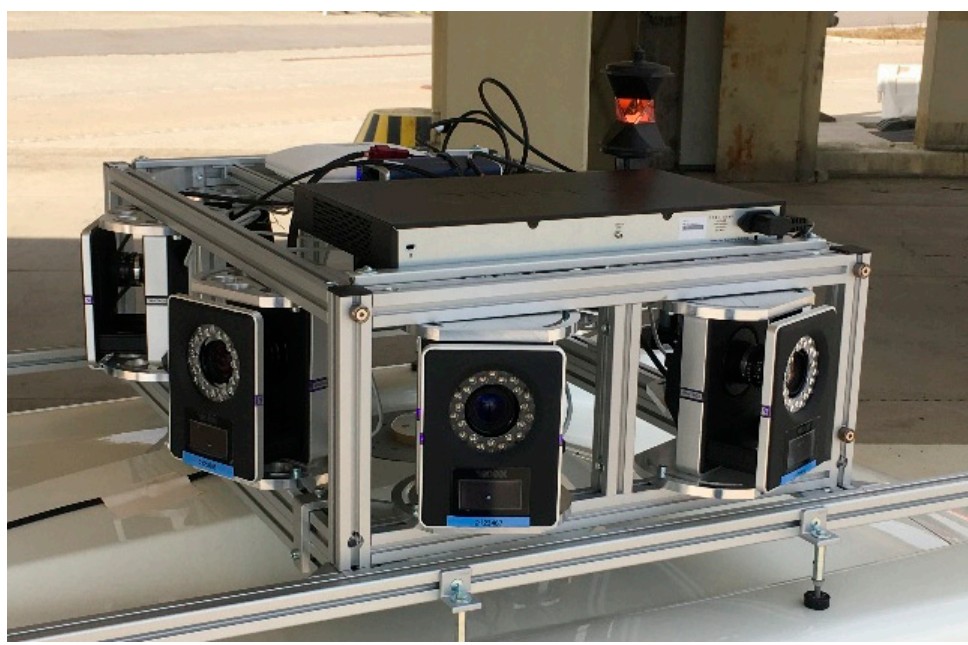

**Figure 1.** Camera rig with 8 Vicon Vantage V5 cameras and one 360° prism.

The accuracy of the pose estimation using this panoramic setup is strongly dependent on the distribution and number of markers. With a Vicon Vantage V5 camera, a 1 pixel shift at a 20 m distance of the marker results in a deviation of 8 mm respective of 0.023° in the measured orientation of the camera. With the measurement of a minimum of 3 markers, the 6 DoF pose can be determined. By measuring more markers, the overdetermination allows for an adjustment and outliers can be detected.

The setup of these sensors brings the following advantages to the system. The motion capture cameras come with integrated preprocessed frames and can reveal the image coordinates of detected markers at a stable frame rate of up to 420 Hz in standard mode [21]. Their well-defined shutter time and ability to simultaneously trigger multiple cameras through a master camera enable accurate synchronization [22]. Installing the cameras onto a mountable rig on top of the car allows the system to be used on different vehicles. The number of cameras used is not dependent on the size of the test site and the energy for the cameras is supplied by the car.

The cheap and light retroreflective markers at the test site can be easily installed on existing infrastructure such as pillars and walls, without any additional construction. Expanding or enhancing a potential test area can be done quickly and the markers are resistant to environmental influences over longer periods of time.

Total stations provide reliable position information by tracking the position of a 360° prism on the camera rig during the initialization phase and the maneuvers. Performing measurements with multiple total stations simultaneously can reduce any systematic errors and allow the monitoring of more complex scenes with partially obstructed areas.

In addition, the robotic total stations can be used during the setup phase to create a reference frame, measure the coordinates of the retroreflective markers, and acquire

positions of additional objects and point clouds. They can also be used to perform the extrinsic calibration between the cameras, the prism, and the reference points on the car.

## 2.1. Physical Setup

The system can be divided into stationary components and those placed on the moving car (Figures 2 and 3) used in the tracking system.

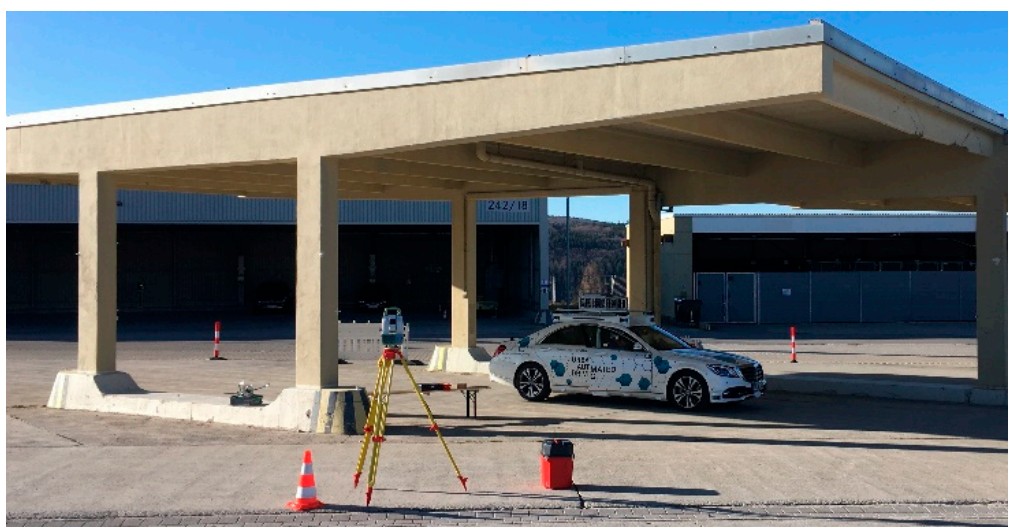

**Figure 2.** Setup of the tracking system at the test site, consisting of one MS60 (on a tripod in the front) and a camera rig on the roof of the car.

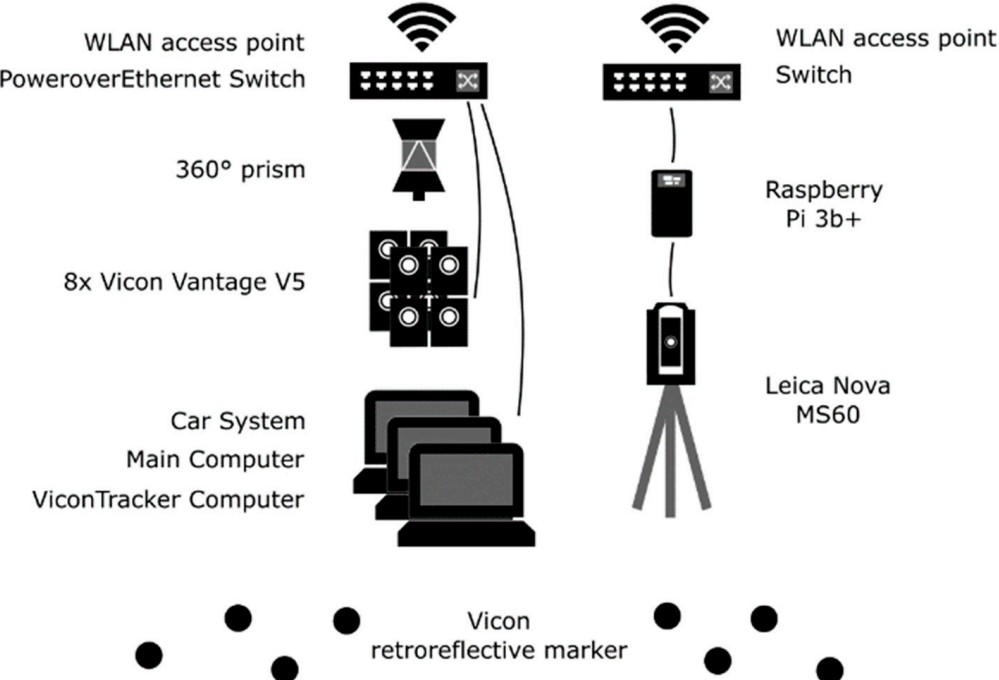

**Figure 3.** Drawing of the components used in the tracking system.

A local world-fixed coordinate frame with physically installed reference points forming a geodetic network is established in the first step. This geodetic network can then be used for a flexible total station setup using the resection method (Figure 4). The retroreflective markers can automatically be measured using the Leica MS60 ATRplus up to a range of 15–25 m, depending on the lighting conditions. To calculate the correct distances, a new prism type must be defined in the total station with an absolute constant of 19.0 mm.

To improve the accuracy and reliability, the markers should be measured from two total station setups. By staking out or re-measuring the markers, their long-term stability can be controlled. Particular attention should be paid to small but systematic errors that cannot be detected during pose estimation, which degrade the results.

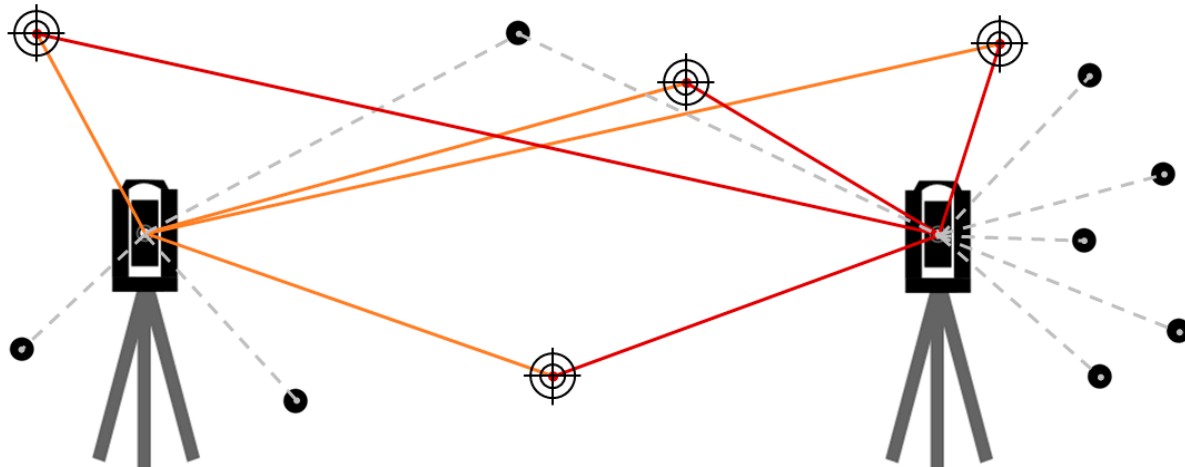

**Figure 4.** Local coordinate system with geodetic reference targets used for setting up the total stations (orange and red lines) to measure the Vicon retroreflective markers (grey dotted lines).

In order to exchange messages with the network on the car, the battery-powered MS60 instruments are connected via TCP/IP to a USB port on a Raspberry Pi 3b+, which is then connected via an Ethernet cable to a switch with a WLAN access point.

All system components placed on the car (except the computers) are attached to a rig (Figure 1). This includes a bolt for mounting the 360° prism, the complementary WLAN access point, and a switch that is powered by the car and supplies the motion capture cameras with Power over Ethernet (PoE). With eight Vicon Vantage 5 cameras, it is possible to achieve a coverage of 95% of the horizon at a distance of 5 m from the car.

This rig forms a coordinate system whose origin is the optical center of the prism, while the definition of a main camera determines the orientation. All cameras (independently and intrinsically calibrated beforehand) as well as the prism are calibrated to each other, so that the system can determine the pose of the rig in relation to the world-fixed coordinate system.

By measuring reference points on the car, a static transformation from the rig to the car-fixed coordinate system is established.

The connection to the vehicle's autonomous system is established over the PoE switch and Ethernet cables, along with the computers needed for controlling the cameras and processing the data.

### 2.2. Logical Setup

A Precision Time Protocol (PTP) grandmaster clock provided by the car's system serves as the basis for the synchronization. The results section describes the procedures that implement accurate timestamps of the measurements to the actual sensor recording time, as well as an analysis of the obtained accuracy.

Communication between the components is performed via ROS nodes, and the master is provided by the car's system.

ROS nodes can be used to subscribe and publish topics, as well as to process the data. All of the published messages include timestamps, and receiving a message from a subscribed topic can be used as a callback to trigger functions. Another method that is used involves loops that are triggered at a predefined rate (or a slower rate if the calculations take longer). In addition, the tf2 package [23] is used to keep track of the different coordinate frames as well as their transformations to each other, which provides buffered information

up to 10 s into the past. Figure 5 shows the information flow, which is next explained in detail.

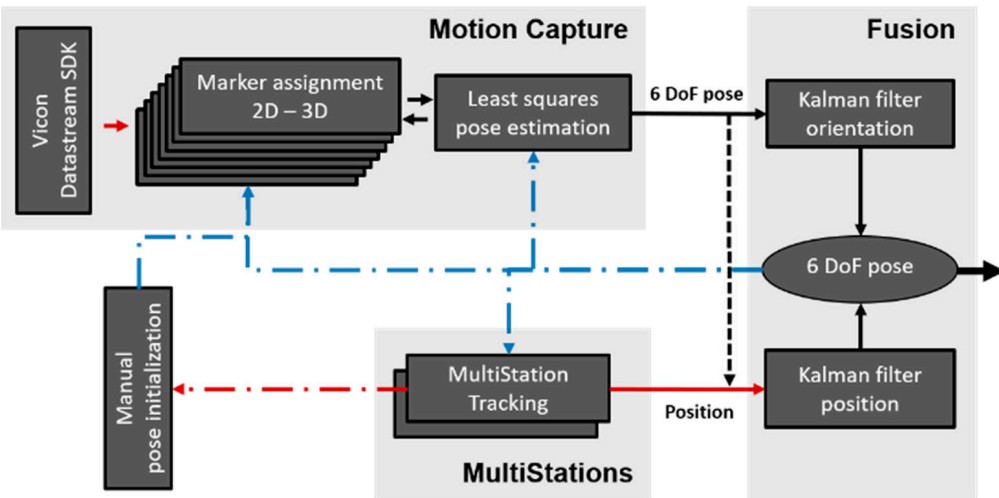

**Figure 5.** The core elements of the logical implementation. Each box represents one ROS node. The measurements are in red and the blue dashed–dotted arrows are pose messages that are needed as initial values.

A script running on the Raspberry Pi triggers the coordinate measurements of the prism at approximately 22 Hz, using the MS60 via the Leica GeoCOM interface. Measurement data are then transmitted back to the Raspberry Pi via the same interface. Based on the measured prism position provided by a total station, other total stations in the network are actively aimed towards the prism's position to enable a fast and automatic lock when the prism enters their field of view.

The measurements—namely coordinates and radii of the markers—provided by the cameras are read out from the Vicon Datastream SDK and are then published as messages into a ROS topic at a rate of 80 Hz.

Upon system startup, the MS60s deliver the prism's position information. With a manually estimated heading, the orientation of the rig is initialized by visually aligning the superimposed projected markers onto the camera images, as shown in Figure 6, where green dots within red circles indicate correctly assigned markers. For circles where the green dot is missing, the marker was not visible or could not be detected. A green dot without a red circle indicates a source of reflectance, which is ignored.

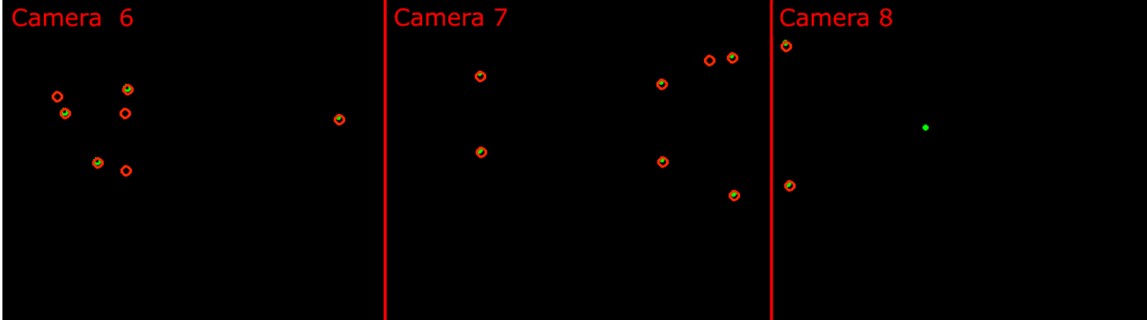

**Figure 6.** Images of three cameras used for pose initialization. Green dots are markers detected by the cameras, while the red circles are projected points based on the pose and the known 3D marker coordinates.

After the initialization, the markers are tracked in the image space by each camera simultaneously. The markers that were previously matched are then explored within a predefined search window. For any unmatched point, a new match is searched for in the image space using the last published pose for the projection. Missing markers are

not a problem if sufficient and well-distributed visible markers remain for robust least squares estimation.

For scenarios with obstructed views, areas of visibility are assigned to the markers so that they are only used for matching if they are visible at that moment in order to avoid any incorrect assignments.

Once all matches of all cameras have been obtained, a least squares estimation for the pose is calculated based on the following formula:

$$K_{C_i}^{-1} * x = [R_b^{C_i} * R_w^b \mid -R_b^{C_i} * R_w^b * \left( r_{w,b}^w + R_w^{b^T} * r_{b,C_i}^b \right)] * X \tag{1}$$

where $X$ represents local coordinates in 3D; $x$ represents image coordinates in 2D; $K$ is the camera matrix; $R$ represents the rotation matrices, i.e., from the starting system (bottom) used to target system (top); $r$ is the calibrated vector between two points; $w$ = world; $b$ = base link; $Ci$ = i-th camera.

This algorithm uses iterations that converge on the optimal solution, which depends on the approximate initial values. These iterations form a bottleneck during near-real-time processing. Therefore, only every second timeframe from the marker tracker is used for the pose estimation (=40 Hz) and the maximum number of iterations is set to three, irrespective of whether the convergence threshold is met. Markers with residuals above a preset threshold are declared as outliers (e.g., caused by incorrect matching or a change in the physical installation), and their matching confidence values are reduced for the following estimations.

Two independent Kalman filters are used to achieve more robust position and orientation determinations. Both are set to run at a rate of 25 Hz. The Kalman filter used for the positioning fuses the measurements from multiple MS60s, and optionally from the calculated Vicon pose (dashed line Figure 5). In contrast, for the orientation, the only source of measurement is the pose estimation. For the orientation, the Kalman filter helps detect incorrectly estimated poses and overcomes short periods of missing pose estimations.

The components of both Kalman filters are combined into the final pose, which can in turn be used as an approximate pose for all other nodes.

Errors introduced by the latency of pose messages are reduced by extrapolating them from the short-term pose history to the current timestamp.

### 2.3. Visualization and Postprocessing

The ROS topics can be recorded into rosbags for postprocessing purposes, such as when replaying recordings, reprocessing the data, or performing additional evaluations such as distance measurement comparisons between the optical tracking system and the sensors on the vehicle. Monitoring of the system status while recording is also possible using a real-time visualization in RViz, which shows the car's movements (Figure 7).

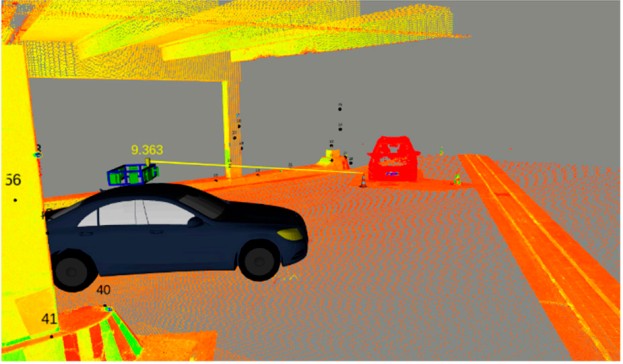

**Figure 7.** Visualization of the point cloud, which represents the static digital twin of the test scene and the moving digital twin of the car.

## 3. Results

### 3.1. Requirement Coverage

The implemented system involving the combination of Leica MS60 MultiStations and Vicon motion capture cameras allowed meet the requirements for industrial applications, as shown in Table 1.

**Table 1.** List of requirements.

| Requirement | Target | Achieved Overall | Achieved by MS60 | Achieved by Vicon | Requirement |
|---|---|---|---|---|---|
| Frequency | >20 Hz | 25 Hz | 20 Hz | 40 Hz | Frequency |
| Latency | 500 ms | 200 ms | 50–60 ms | 40 ms | Latency |
| Positional error | 5 mm (3σ) | 5 mm | 5 mm | 5–30 mm | Positional error |
| Orientational error | 0.3° (3σ) | 0.03° | - | 0.003–0.3° | Orientational error |

In addition to these numerical requirements, synchronization using PTP and full control of the system from within the test vehicle were achieved.

### 3.2. Synchronization

The synchronization of all sensors creates a basis for accurate spatiotemporal data. The required accuracy of the synchronization depends on the maximal velocity of the object and the targeted position accuracy, as shown in Table 2.

**Table 2.** Relationships between velocity, synchronization, and accuracy.

| Velocity | Targeted Position Accuracy @ 1σ | Requirement for Maximal Synchronization Offset |
|---|---|---|
| 1 m/s | 1 mm | 1 ms |
| 1 m/s | 5 mm | 5 ms |
| 5 m/s | 5 mm | 1 ms |

The internal synchronization of the car is established via a grandmaster clock that communicates over PTP. The same grandmaster clock is used for synchronizing the optical tracking system computers in software stamping mode. The following sections describe the additional procedures needed to synchronize the sensors' time systems to PTP.

The motion capture data streamed from the Datastream SDK show a latency of around 40 ms. For the camera synchronization, the UDP trigger signals sent from the main camera are used to connect the frame numbers of the motion capture system to the PTP timestamp. An additional offset dependent on the frame rate is applied in order to consider the offset between the UDP trigger signal and the actual shutter opening time. A frame rate of 80 Hz corresponds to 11.5 ms.

For the MS60 instruments, the PTP synchronization over WLAN between the Raspberry Pi slave (using the PTP implementation ptpd2) and the grandmaster clock is investigated. Over a period of multiple hours, the analysis shows a mean offset from the master clock of 0.01 ms with a 3σ standard deviation of 1 ms (Figure 8).

The link between the timestamps as delivered by the MS60 measurements to the PTP time is established through the estimation of a clock model with a scale and offset:

$$t_{PTP} = t_{MS} * m + t_0 + t_c \tag{2}$$

where $t_{PTP}$ is the ROS time, $t_{MS}$ is the MS60-time, $m$ is the time scale, $t_0$ is the offset, and $t_c$ is the constant offset for numerical reasons.

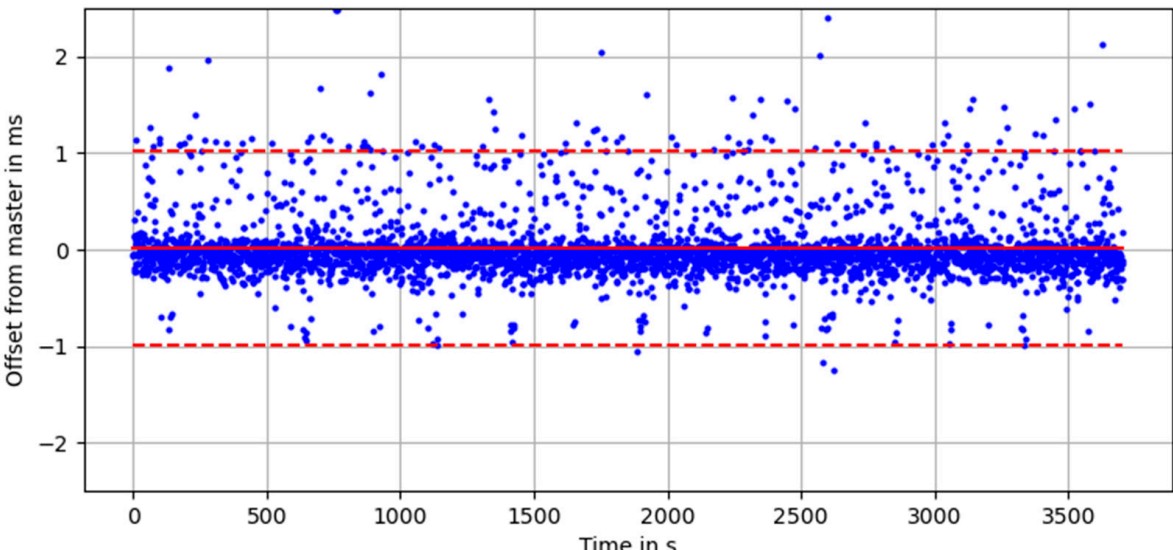

**Figure 8.** Time offsets of a slave to its master over WLAN.

To query the MS60 sensor time, a dedicated GeoCOM command that takes approximately 10 ms between request and reception (typical values for other requests have a duration above 40 ms) delivers the observations used for calculating the time offset of the MS60. The return time is taken from the middle third of the entire timespan, so an unknown offset of 3.3 ms remains.

At startup, the clock model is initialized with 180 samples of the sensor time. During the tracking of the prism, a new sample is taken every three seconds that is weighted against the time span between sending and receiving. Every 30 s, the parameters m and $t_0$ are estimated again. Investigations show that these sparse updates are more than sufficient, and even after 10 min without an update no significant drift of the clock model appears.

The reliability of the continuous synchronization is controlled by measuring independent samples every five seconds and comparing their deviations to the clock model. The results from two hours of measurements show a mean shift of $-0.03$ and a $3\sigma$ standard deviation of 1.02 ms.

In summary, the synchronization of the MS60 instruments has empirically determined standard deviation ($3\sigma$) values of 1 ms due to the WLAN usage and 1.02 ms due to random scatter. Due to the unknown measurement time, an unknown offset estimated to be in the range of 3.3 ms remains.

### 3.3. MS60 Position Accuracy

Measurements from two MS60 instruments onto one moving prism are used to quantify the velocity-dependent influence of the synchronization and the automatic target recognition (ATRplus) based on the coordinate accuracy. Other potential error sources are reduced by positioning the MS60 instruments close to each other—resulting in a minimal influence of the orientation of the 360° prism—and using the same fixpoints for the resection. The measured trajectories are smooth movements over an area of 15 × 50 m, with measurement distances of up to 50 m tracked at a rate of 20 Hz.

Table 3 shows a comparison between the two trajectories, where for each measurement of one MS60, the corresponding coordinates of the second trajectory are determined via a linear interpolation over time.

**Table 3.** Coordinate differences dependent on the velocity.

| Velocity | Mean of Coordinate Differences in mm | Standard Deviation of Single Measure in mm | Maximal Difference in mm | Samples |
|---|---|---|---|---|
| 0–1 m/s | 2.0 | 0.4 | 4.5 | 2281 |
| 1–2 m/s | 4.2 | 3.9 | 22.2 | 1782 |
| 2–3 m/s | 7.2 | 5.5 | 44.7 | 2062 |
| 3–4 m/s | 8.9 | 7.7 | 55.4 | 781 |
| 4–5 m/s | 8.8 | 7.4 | 64.1 | 620 |

These results are influenced by the errors occurring on both MS60 instruments and show the expected result of increased differences for higher velocities. For these higher velocities, some of the deviations are caused by the linear interpolation of a curved trajectory.

These results show that the tracking functionality of MS60 instruments only meets the requirements at slower speeds—as they appear in the proposed application area with near-range maneuvers—and that the standard deviation of the differences is very small compared to the mean value.

*3.4. Comparison of Position Accuracy of Subsystems*

The measured position of the system's reference point located at the prism center can be determined independently by multiple MS60 instruments and from the 6DoF pose estimation from the motion capture system. A comparison of these positions can be used to check single measurements and to detect continuous offsets caused by erroneous setups or calibrations.

Figure 9 shows a seven second excerpt of a maneuver performed in the laboratory on a moving table. In Figure 10, a more detailed view of the same maneuver in the encircled region is shown. This indicates a small spike from the motion capture pose estimation that can be cleaned up in a postprocessing step, as well as a dynamic offset between the two MS60 instruments that can be explained through cyclic deviations introduced by the 360° prism.

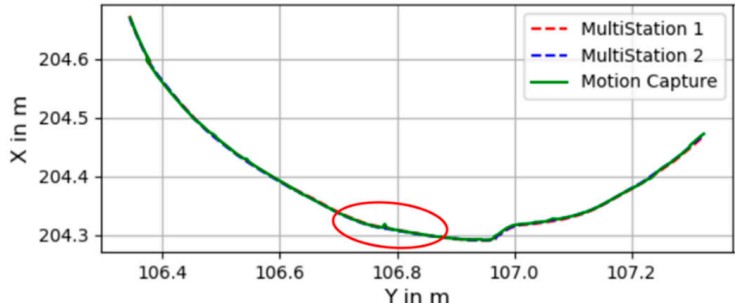

**Figure 9.** Measurements of circular rotation.

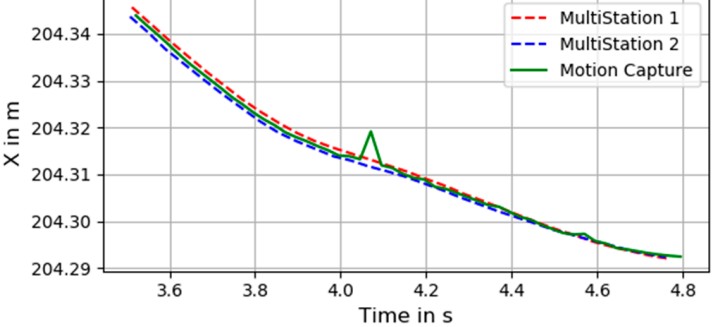

**Figure 10.** Detailed view of the encircled region shown in Figure 9.

### 3.5. Accuracy of the Motion Capture System

The resulting pose estimations are controlled with an AT401 laser tracker with 3D point accuracy in the range of ±15 μm + 6 μm. For these checks, the system is set up in different poses within a laboratory environment with 20–25 retroreflective markers in the motion capture camera's field of view. The GPR121 prism serves as the main reference point and three points on the rig, where a Red Ring Reflector 1.5 can be placed, form the basis of a coordinate system (Figure 11). All four points are measured by the laser tracker.

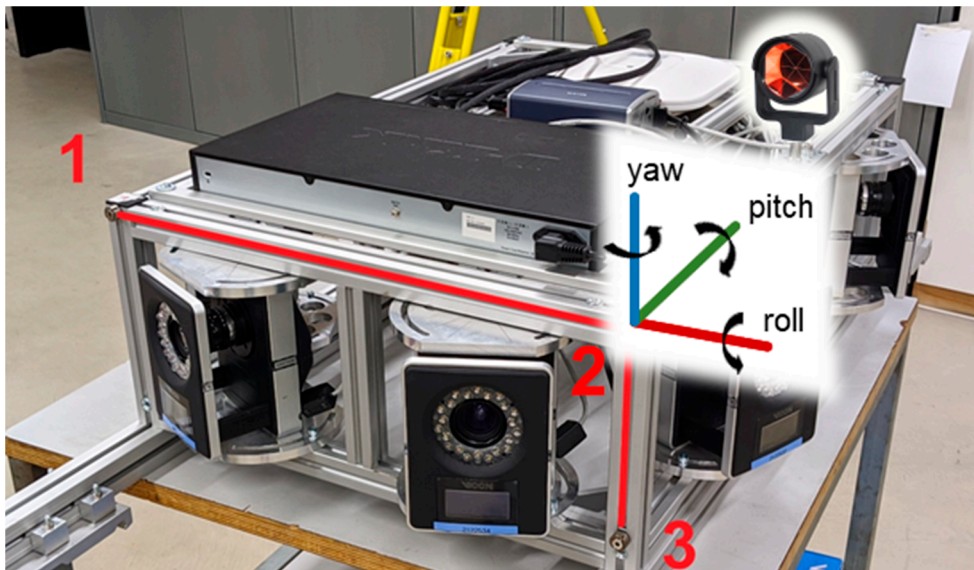

**Figure 11.** Points measured with the laser tracker. Points 1 to 3 are used for orientation, while the prism is used for positioning.

As listed in Table 4, the resulting standard deviations of these static measurements show a clear achievement of the requirements under these laboratory conditions.

**Table 4.** Standard deviations resulting from six pose measurements.

| | Roll in Deg | Pitch in Deg | Yaw in Deg | X in mm | Y in mm | Z in mm |
|---|---|---|---|---|---|---|
| Standard deviation (1σ) | 0.111 | 0.100 | 0.077 | 0.429 | 0.374 | 0.273 |

The smaller standard deviation around the *z*-axis (yaw) can be explained by the configuration of the cameras distributed in a panoramic view around that axis, whereas the reliability around the other axes is less strongly controlled.

## 4. Discussion

By using the optical tracking system, a vehicle's sensory and algorithmic output can be evaluated in real time during the execution of near-range maneuvers. This is achieved by visualizing the algorithmic output and the car's reference pose on the onboard display unit or on any laptop connected to the vehicle. In addition, the data can be recorded in rosbag files for later in-depth investigation and validation of the autonomous vehicle.

To achieve the given requirements, every step must be performed at the highest level of accuracy. This includes establishing an accurate and stable fixed point network; controlling the distribution, measurement, and maintenance of the retroreflective markers; and calibrating and synchronizing all sensors.

Because of the high accuracy requirements, this process is time-consuming and requires high-quality equipment, making the entire system expensive. Compromises to the

accuracy requirements could simplify the system and make it less expensive. Environments with shiny surfaces causing interfering reflections can make it challenging to configure the motion capture cameras. Locations that do not contain infrastructure on all sides where markers can be placed are also problematic for reliable pose estimations.

To further improve the optical tracking system, the following options could be considered:

- An improvement of the MS60s' measurements could be achieved through enhanced synchronization. However, this would require changes to the MS60s' firmware or even the hardware;
- As suggested in [24], a correction process to mitigate the systematic cyclic deviations of the 360° prism can be pursued, since the alignment of the prism towards the MS60s is always known;
- In addition, supporting the orientation determination through the integration of an IMU into the existing system could help bridge the poor short-term geometric distribution of the markers and make the marker detection and assignment process more robust.

## 5. Conclusions

The proposed optical tracking system provides an integrated workflow that can be used to prepare a digital twin of a test scene. It uses geodetic total stations and a camera-based motion capture system for highly accurate 6DoF pose estimation as a reference for the validation of an autonomous driving vehicle in near-range scenarios.

The system can cover test drive distance range of up to 100 m, which is sufficient for most parking scenarios and environments. The requirements of a 6 degree of freedom (DoF) pose with a position uncertainty below 5 mm ($3\sigma$), orientation uncertainty below $0.3°$ ($3\sigma$), at a frequency higher than 20 Hz, and with a latency smaller than 500 ms can be met. The reference pose is published as messages into an ROS topic.

**Author Contributions:** Conceptualization, U.K., L.S. and D.E.G.; data curation, U.K., L.S. and D.E.G.; formal analysis, U.K. and D.E.G.; funding acquisition, A.W.; investigation, U.K.; methodology, U.K. and D.E.G.; project administration, L.S.; resources, L.S. and A.W.; software, U.K.; supervision, D.E.G.; validation, U.K., L.S., D.E.G. and A.W.; visualization, U.K.; writing—original draft, U.K., L.S., D.E.G. and A.W.; writing—review and editing, D.E.G. and U.K. All authors have read and agreed to the published version of the manuscript.

**Funding:** This research received no external funding.

**Institutional Review Board Statement:** Not applicable.

**Informed Consent Statement:** Not applicable.

**Data Availability Statement:** Data sharing is not applicable to this article.

**Conflicts of Interest:** The authors declare no conflict of interest.

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
