# Peer review of "Highly Accurate Pose Estimation as a Reference for Autonomous Vehicles in Near-Range Scenarios"

_remotesensing, doi:10.3390/rs14010090_

Round 1

Reviewer 1 Report

The paper addresses a novel solution that can be applied to autonomous vehicles for low-speed maneuvers.

As presented, the solution is like the methods used for shape inspection using cameras. In this case, markers are required to identify the 3D shape of the surroundings correctly.

From this point of view, it means that special attention should be paid to the infrastructure for the method to be fully functional. This adds extra constraints for the implementation of the proposed solution as it addresses the vehicle itself and the infrastructure.

What will happen in case of missing markers or poorly maintained ones?

Although the method proposed is clearly presented and exemplified, the authors should present the shortcomings of this research. A critical discussion is required, and they should indicate the possible solutions for navigation when the markers are missing.

To accurately notify the reader about the current finding, the authors should add a line in the Abstract briefly describing the method and prerequisites.

Author Response

Dear reviewer,

thank you very much for your valuable Comments and Suggestions. We have tried to implement them in our paper. Below you will find our answers to the individual findings in red.

The paper addresses a novel solution that can be applied to autonomous vehicles for low-speed maneuvers.

As presented, the solution is like the methods used for shape inspection using cameras. In this case, markers are required to identify the 3D shape of the surroundings correctly.

From this point of view, it means that special attention should be paid to the infrastructure for the method to be fully functional. This adds extra constraints for the implementation of the proposed solution as it addresses the vehicle itself and the infrastructure.

We addressed this points in the discussion.

What will happen in case of missing markers or poorly maintained ones?

We added the following statement: Missing markers are not a problem as long as sufficient and well-distributed visible markers remain for robust least squares estimation.

As described, can outliers (e.g. caused by a bigger shift of the physical markers) be detected in the least-squares estimation. But to reach the highest accuracies it is important to maintenance such as cleaning the markers, or remeasuring them after some time.

Although the method proposed is clearly presented and exemplified, the authors should present the shortcomings of this research. A critical discussion is required, and they should indicate the possible solutions for navigation when the markers are missing.

Added problematic of interfering reflections (caused by motion capture system method of passive markers).

Changed statement for IMU: short-term bridging of missing/poor marker distribution

We addressed this points in the discussion.

To accurately notify the reader about the current finding, the authors should add a line in the Abstract briefly describing the method and prerequisites.

Abstract slightly improved to give insight into method.

Reviewer 2 Report

The reviewed article contains interesting considerations. The extensive practical analysis of issues related to the highly accurate pose estimation as reference for autonomous vehicles in near-range scenarios, but the current version need improved, for details se below:

  • article title: according style guide, nouns, pronouns, verbs, adjectives, and adverbs are the words capitalized in titles of articles,
  • line 4, change “Kälin1” to “Kälin 1”,
  • line 4, change “Staffa2” to “Staffa 2”,
  • line 4, change “Grimm1,*” to “Grimm 1,*”,
  • line 4, change “Wendt2” to “Wendt 2”,
  • line 24-26, left column: give them the name of the author and the title of the article,
  • add the extended state of the issue (the authors need to explain the difference between the current study and the available literature more precisely; use publications not older than 3-4 years),
  • line 329, change “?0” to “?0”,
  • line 429-435: use initials instead of names and surnames,
  • References: is required Abbreviated Journal Name,
  • References: include the digital object identifier (DOI) for all references where available,
  • References: please format according to the journal guidelines
    https://www.mdpi.com/files/word-templates/remotesensing-template.dot
  • References: a very small collection of bibliographies.

In future publications, authors should devote more time to editing the article according to the requirements of the journal. This will then avoid quite a number of insights into editing.

The reviewed article is a valuable publication. It can serve readers as a set of knowledge that can be used as a basis for further innovative and implementation studies.

Author Response

Dear reviewer,

thank you very much for your valuable Comments and Suggestions. We have tried to implement them in our paper. Below you will find our answers to the individual findings in red.

The reviewed article contains interesting considerations. The extensive practical analysis of issues related to the highly accurate pose estimation as reference for autonomous vehicles in near-range scenarios, but the current version need improved, for details se below:

  • article title: according style guide, nouns, pronouns, verbs, adjectives, and adverbs are the words capitalized in titles of articles,
  • line 4, change “Kälin1” to “Kälin 1”,
  • line 4, change “Staffa2” to “Staffa 2”,
  • line 4, change “Grimm1,*” to “Grimm 1,*”,
  • line 4, change “Wendt2” to “Wendt 2”,
  • line 24-26, left column: give them the name of the author and the title of the article,

made all changes as suggested

  • add the extended state of the issue (the authors need to explain the difference between the current study and the available literature more precisely; use publications not older than 3-4 years),

we added one additional study from 2019 presenting a different approach for a similar issue.

  • line 329, change “?0” to “?0”,
  • line 429-435: use initials instead of names and surnames,
  • References: is required Abbreviated Journal Name,
  • References: include the digital object identifier (DOI) for all references where available,
  • References: please format according to the journal guidelines
    https://www.mdpi.com/files/word-templates/remotesensing-template.dot
  • References: a very small collection of bibliographies.

made all changes as suggested

In future publications, authors should devote more time to editing the article according to the requirements of the journal. This will then avoid quite a number of insights into editing.

The reviewed article is a valuable publication. It can serve readers as a set of knowledge that can be used as a basis for further innovative and implementation studies.

Thank you very much!

Reviewer 3 Report

The manuscript proposes 
1. a method to validate the accuracy and reliability of onboard sensors for object detection and localization in driver assistance
2. a novel tracking system.

In general, I think the novelty of the proposed method is somehow limited; however, I think the described system is absolutely valuable for potential readers. Please find comments below.

- Contributions and Methodology.
a.From Figure 4, although It's clear that you use Kalman filter to recover 6DOF of car. You should add a diagram to describe the data processing flow more clearly, such as establishing a (local) world-fixed coordinate and getting the coordinates of Multistations and all markers. 
There are redundant observations. One Leica station can observe multiple markers, and multiple Leica stations can keep one tag.
b.You can briefly introduce how you use an 8-lens panoramic camera to obtain the coordinates of all markers.

-- Experimental evaluations.
You can add a set of experiments using an 8-lens panoramic camera and a 6DOF of the car to recover some markers' (local) world-fixed coordinate and then compare them with the actual value. Measure the error of the whole system.

-Chart 
a. In Figure 5. Maybe you don't have to keep the whole software interface. It's better to keep only the window of the point cloud.
b. You should revise Figure 6 to make it clear and professional. It seems to be a software screenshot.

Author Response

Dear reviewer,

thank you very much for your valuable Comments and Suggestions. We have tried to implement them in our paper. Below you will find our answers to the individual findings in red.

The manuscript proposes 
1. a method to validate the accuracy and reliability of onboard sensors for object detection and localization in driver assistance
2. a novel tracking system.

In general, I think the novelty of the proposed method is somehow limited; however, I think the described system is absolutely valuable for potential readers. Please find comments below.

- Contributions and Methodology.
a.From Figure 4, although It's clear that you use Kalman filter to recover 6DOF of car. You should add a diagram to describe the data processing flow more clearly, such as establishing a (local) world-fixed coordinate and getting the coordinates of Multistations and all markers. 

There are redundant observations. One Leica station can observe multiple markers, and multiple Leica stations can keep one tag. To explain the surveying of the local reference frame and markers an additional Figure 3 is introduced, and the corresponding section is expanded. Included is now also the note on how to control the markers by stake-out and if needed perform new measurements.

b.You can briefly introduce how you use an 8-lens panoramic camera to obtain the coordinates of all markers.

The 3D world coordinates of the markers are determined using the total station(s) as now shown in the newly added Figure 3. To initialize the marker detection, a first manual orientation must be done, which is now illustrated by the newly added Figure 6.

-- Experimental evaluations.
You can add a set of experiments using an 8-lens panoramic camera and a 6DOF of the car to recover some markers' (local) world-fixed coordinate and then compare them with the actual value. Measure the error of the whole system.

We included the results of another set of measurements in section 3.5 where the orientation and also position was checked with the laser tracker. These measurements were done before remeasuring all markers with a totals station and therefore the values for the orientation are slightly worse. However, to keep all values consistent and because the orientation values are still within the tolerance, we decided to use these values for all 6DOF accuracy investigations.

-Chart 
a. In Figure 5. Maybe you don't have to keep the whole software interface. It's better to keep only the window of the point cloud.
b. You should revise Figure 6 to make it clear and professional. It seems to be a software screenshot.

These figures have been adapted to highlight the message and fit better into the layout.

Round 2

Reviewer 3 Report

The authors have made satisfactory responses to my previous concerns.